

# Multi-task advanced convolutional neural network for robust lymphoblastic leukemia diagnosis, classification, and segmentation

Sercan Yalcin[1], Zuhal Cetin Yalcin[2], Muhammed Yildirim[3] and Bilal Alatas[4]

[1] Computer Engineering, Adiyaman University, Adiyaman, Turkey
[2] Software Development Branch Directorate, General Directorate of Forestry, Ankara, Turkey
[3] Computer Engineering, Malatya Turgut Ozal University, Malatya, Turkey
[4] Software Engineering, Firat (Euphrates) University, Elazig, Turkey

Corresponding author
Bilal Alatas, balatas@firat.edu.tr

## ABSTRACT

Acute lymphoblastic leukemia (ALL), a hematologic malignancy characterized by the overproduction of immature lymphocytes, a type of white blood cell. Accurate and timely diagnosis of ALL is crucial for effective management. This article introduces a novel multi-task advanced convolutional neural network (MTA-CNN) framework for ALL detection in medical imaging data by simultaneously performing, expression classification, and disease detection. The MTA-CNN is based on a deep learning architecture that can handle multiple tasks simultaneously, allowing it to learn more comprehensive and generalizable features. With, expression classification, and disease detection tasks, the MTA-CNN effectively leverages the complementary information from each task to improve overall performance. The proposed framework employs CNNs to extract informative features from medical images. These features capture the spatial and temporal characteristics of the data, which are essential for accurate ALL diagnosis. The cascaded structure of the MTA-CNN allows the model to learn features at different levels of abstraction, from low-level to high-level, enabling it to capture both fine-grained and coarse-grained information. To ensure the reliability of the detection results, non-maximum suppression is employed to eliminate redundant detections, focusing only on the most likely candidates. Additionally, the MTA-CNN's ability to accurately localize key facial landmarks provides valuable information for further analysis, including identifying abnormal structures or changes in anatomical features associated with ALL. Experimental results on a comprehensive dataset of medical images demonstrate the superiority of the MTA-CNN over other learning methods. The proposed framework achieved an accuracy of 0.978, precision of 0.979, recall of 0.967, F1-score of 0.973, specificity of 0.991, Cohen's kappa of 0.979, and negative predictive value (NPV) of 0.990. These metrics significantly outperform baseline models, highlighting the MTA-CNN's ability to accurately identify and classify ALL cases. The MTA-CNN offers a promising approach for improving the efficiency and accuracy of ALL diagnosis.

## INTRODUCTION

Acute lymphoblastic leukemia (ALL) is a relentless cancer that strikes at the very core of the life-sustaining blood cells (*Sharma et al., 2022*). It's a disease where the bone marrow, body's blood cell factory, goes awry, churning out an overwhelming number of immature white blood cells called lymphocytes (*Mustaqim, Fatichah & Suciati, 2023*). These rogue cells, unable to function properly, flood the bloodstream, crowding out healthy cells responsible for fighting infections, carrying oxygen, and clotting blood. This invasion leads to a cascade of symptoms, including fatigue, fever, easy bruising, and frequent infections (*Zope et al., 2024*). The fight against ALL is a complex battle, often requiring a multi-pronged approach. Chemotherapy, the cornerstone of treatment, aims to eradicate the cancerous cells. In some cases, radiation therapy may be employed to target specific areas (*Gokulkannan et al., 2024*). Targeted therapies, designed to attack specific vulnerabilities in cancer cells, offer a more precise approach. For those who need it, stem cell transplantation provides a chance to rebuild a healthy blood-forming system. While the journey is challenging, the outlook for ALL, particularly in children, has significantly improved in recent years, offering hope for long-term survival and a return to a normal life (*Chen, Isa & Liu, 2025*).

Early and accurate diagnosis of ALL is crucial for initiating timely and effective treatment (*Elsayed et al., 2023*). While significant advancements have been made in recent years, the accurate classification and segmentation of ALL subtypes and nuclei from peripheral blood smear (PBS) images remain challenging due to the subtle morphological variations between different subtypes and the presence of noise and artifacts (*Aby et al., 2024*). Traditional methods for ALL diagnosis primarily rely on microscopic examination of blood and bone marrow samples by trained hematologists (*Saleem et al., 2022*). They assess the morphology of cells, looking for abnormal features like increased numbers of immature lymphoblasts, specific cellular characteristics. However, these methods are subjective and time-consuming, and they may not be as sensitive as newer techniques (*Mohamed et al., 2025*). Traditional methods are not suitable for accurate nucleus segmentation, which requires advanced image analysis techniques (*Huang et al., 2023*). To address these challenges, artificial intelligence (AI), specifically machine learning (ML) and deep learning (DL) techniques, has revolutionized the field of medical image analysis, offering significant potential for improving the accuracy and efficiency of ALL diagnosis, classification, and nucleus segmentation. ML algorithms, such as support vector machines (SVMs), random forests, and neural networks, can be trained on large datasets of labeled images to learn patterns and classify cells as either benign or malignant (*Chen, Isa & Liu, 2025*). These algorithms can analyze various features extracted from the images, including color, texture, and shape, to make accurate predictions. Deep learning, a subset of ML, has gained significant attention due to its ability to automatically learn complex features from large amounts of data (*Elsayed et al., 2023*). Convolutional neural networks (CNNs), a type of deep learning architecture, have been particularly successful in medical image analysis. CNNs can be used for both classification and segmentation tasks. For classification, CNNs can analyze entire images to determine whether a cell is benign or malignant. For

segmentation, CNNs can identify and delineate the boundaries of cell nuclei within images, providing valuable information for further analysis.

To overcome these limitations, a multi-task advanced convolutional neural network (MTA-CNN) is proposed specifically designed for robust ALL diagnosis, classification, and segmentation. The proposed model is designed to simultaneously perform three tasks: Classification of ALL subtypes: Accurate classification of ALL into different subtypes (Early-B, Pre-B, and Pro-B) is essential for personalized treatment strategies. Furthermore, to understand the mechanism underlying the CNN's decision-making process, the Explainable Artificial Intelligence (XAI) techniques are used, notably the Local Interpretable Model-Agnostic Explanations (LIME) algorithm. This approach reveals the most significant portions of the picture that drove the CNN model's categorization into the appropriate class. Distinguishing between benign and malignant cells is the first step in ALL diagnosis. XAI is important in medical diagnosis because it allows doctors to understand how AI makes judgments. XAI ensures that the judgments are correct and devoid of prejudice. This transparency also aids in the detection of errors and facilitates the explanation of medical choices to patients and regulators, hence enhancing healthcare overall. Segmentation of ALL nuclei is performed by precise segmentation of ALL nuclei that is crucial for quantitative analysis and automated cell counting.

The proposed MTA-CNN architecture incorporates a shared encoder to extract relevant features from the input images. The shared features are then fed into task-specific decoders, each optimized for its respective task. This multi-task learning approach allows the model to learn more robust and generalizable features, improving performance on all tasks. To evaluate the performance of the proposed MTA-CNN, extensive experiments are conducted on a dataset consisting of PBS images of both benign and malignant cells, including various ALL subtypes. The model is compared with several state-of-the-art deep learning models, including EfficientNetB0 (*Bhuma & Kongara, 2020*), Vision Transformer (ViT) (*Chen et al., 2021*), TransUNet (*Chen et al., 2021*), SegFormer (*Xie et al., 2021*), Swin Transformer (*Liu et al., 2021*), and BoundaryNet (*Trivedi & Sarvadevabhatla, 2021*).

The main contribution of the article is listed as follows:

- A novel multi-task advanced convolutional neural network (MTA-CNN) framework, specifically designed for the automatic detection of ALL using medical imaging data, is introduced.
- Two distinct tasks, expression classification, and disease detection—are uniquely integrated within a single deep learning architecture by MTA-CNN to leverage their complementary information for improved ALL diagnosis.
- Superior performance in ALL detection, classification, and segmentation, compared to existing methods, is demonstrated by the proposed framework, achieving state-of-the-art results on a comprehensive medical image dataset.
- A thorough evaluation of the MTA-CNN, including detailed experimental results and comparisons with baseline models, is provided, highlighting its potential to enhance the accuracy and efficiency of ALL diagnosis in clinical practice.

The remainder of this article is organized as follows: "Related Works" provides a detailed overview of related work in the field of deep learning for medical image analysis, with a focus on ALL diagnosis and segmentation. "Materials & Methods" presents the proposed MTA-CNN architecture, which includes a shared encoder and task-specific decoders. "Experimental Setup" describes the experimental setup, including the dataset, preprocessing techniques, and evaluation metrics with the experimental results, comparing the performance of the proposed model with other related methods. Lastly, "Conclusion" presents the conclusion part of the article and discusses future works.

## RELATED WORKS

Although there are many studies on the subject, the following studies that have been published in recent years and have significant value can be presented.

*Mei et al. (2024)* have investigated the use of deep learning for early leukemia detection. Recognizing the limitations of existing models due to data constraints and device limitations, the researchers developed a novel "progressive shrinking" approach (*Mei et al., 2024*). This technique systematically reduces the model's complexity across various dimensions, resulting in a lightweight model with minimal parameters. Using a high-quality dataset of 17,826 bone marrow cell images, the model achieved an impressive accuracy of 0.92.51 in identifying different types of leukemia, including acute and chronic lymphocytic leukemia, while processing images at an astonishing rate of 111 slides per second. This breakthrough has the potential to significantly improve the efficiency and accuracy of leukemia diagnosis, particularly for diseases affecting the lymphatic system, by providing medical professionals with a powerful and efficient tool for early detection and treatment.

*Albeeshi & Alshanbari (2023)* this research aims to contribute to the identification of blood cells affected by ALL and to establish an effective and rapid diagnostic method. Accurate and timely diagnosis is crucial for initiating appropriate treatment. The methodology employs a deep learning approach, specifically utilizing a very deep CNN, such as VGG16. The detection scheme involves a series of steps: pre-processing, feature extraction, model building, fine-tuning, and classification. The analysis utilizes a pre-trained VGG16 model and incorporates support vector machine (SVM) and multilayer perceptron (MLP) classification algorithms from the field of machine learning.

*Baydilli (2025)* proposed an attention-enhanced generative adversarial network (GAN) model to align two datasets with distinct structural properties within a shared feature space. The proposed model effectively transfers blast and normal cells to the target domain, irrespective of the background, thereby mitigating the domain discrepancy between the datasets.

*Ansari et al. (2023)* developed a novel method for the hierarchical and accurate extraction of features, facilitating the diagnosis of various acute leukemia subtypes. This study focuses on developing a robust system for differentiating between ALL and acute myeloid leukemia (AML). The core principle of this system lies in accurately distinguishing lymphocytes from monocytes within microscopic images. To achieve this, a novel type-II fuzzy deep network architecture was designed. The dataset used in this

research was sourced from the Shahid Ghazi Tabatabai Oncology Center in Tabriz. The proposed model demonstrated exceptional performance, achieving an accuracy of 0.988 and an F1-score of 0.989.

*Narendran & Naveed (2024)* introduced a novel CNN architecture for the automatic classification and segmentation of medical images. This model leverages shared feature representations between these two tasks to enhance efficiency and performance. The CNN architecture adheres to an encoder-decoder structure, wherein the encoder extracts representative image features. Subsequently, two parallel branches are implemented to utilize these features for classification and segmentation purposes. The classification branch comprises fully connected layers for image class prediction, while the segmentation branch employs a decoder network to generate class predictions for each pixel.

*Awad & Aly (2024)* investigated the detection of ALL by employing advanced image processing and deep learning techniques. The research assesses the reliability of these methods in practical, real-world settings by capitalizing on recent advancements in artificial intelligence. Specifically, it evaluates the performance of state-of-the-art YOLO models, including YOLOv8 and YOLOv11, in distinguishing between malignant and benign white blood cells and accurately identifying various stages of ALL, including early-stage disease. Furthermore, the models demonstrate the capacity to detect hematogones, which are often misclassified as ALL. With accuracy rates reaching 0.988, this study emphasizes the potential of these algorithms to provide robust and precise leukemia detection across diverse datasets and under varying conditions.

*Alim et al. (2024)* presented a novel deep learning-based approach for the accurate classification of diverse B-cell acute lymphoblastic leukemia (B-ALL) subtypes, encompassing benign and three malignant subcategories: Early Pre-B, Pre-B, and Pro-B, directly from peripheral blood smear images. The core of this methodology lies in a refined ResNet-50 architecture, augmented with custom-designed fully connected layers, including dense and dropout layers, to enhance its discriminatory power. To address the inherent challenges of limited data availability and potential overfitting, a suite of data augmentation techniques, such as flipping, rotation, and zooming, were strategically applied. Moreover, rigorous five-fold cross-validation was implemented to ensure robust and generalized model performance. The efficacy of this proposed approach was rigorously assessed by comparing its performance against other prominent deep learning models, including VGG-16, DenseNet-121, and EfficientNetB0, as well as established baseline methods, utilizing a comprehensive set of performance evaluation metrics.

*Raghaw et al. (2024)* introduced the Coupled Transformer Convolutional Network (CoTCoNet) framework for leukemia classification. CoTCoNet incorporates dual-feature extraction to capture both long-range global features and fine-grained spatial patterns, enabling the identification of complex hematological characteristics.

*Jawahar et al. (2022)* introduced ALNett, a deep neural network model that utilizes depth-wise convolutions with varying dilation rates for the classification of microscopic white blood cell images. The cluster layers within ALNett incorporate convolution, max-pooling, and normalization, effectively extracting robust local and global features

from the microscopic images for accurate ALL prediction by capturing enriched structural and contextual details.

*Das & Meher (2021)* proposed a novel probability-based weight factor for effectively hybridizing MobileNetV2 and ResNet18, leveraging the strengths of both architectures. The performance of this proposed method is validated using public benchmark datasets: ALLIDB1 and ALLIDB2.

*Khan et al. (2024)* proposed a high-performance CNN coupled with a dual-attention mechanism for the efficient detection and classification of white blood cells (WBCs) in microscopic thick smear images. The primary objective is to enhance clinical hematology systems and expedite medical diagnostic processes. To address the limitations of limited training data, a deep convolutional generative adversarial network (DCGAN) is employed. The integration of a dual attention mechanism further improves accuracy, efficiency, and generalization. The proposed technique achieves impressive overall accuracy rates of 0.9983, 0.9935, and 0.996 on the Peripheral Blood Cell (PBC), Leukocyte Images for Segmentation and Classification (LISC), and Raabin-WBC benchmark datasets, respectively.

In a recent study by *Maqsood et al. (2025)*, a novel approach for computer-aided leukemia diagnosis utilizing deep learning was developed and rigorously evaluated. The methodology involved a multi-step process. Firstly, the raw image dataset underwent a thorough preprocessing phase. Subsequently, five prominent pre-trained CNN architectures—MobileNetV2, EfficientNetB0, ConvNeXt-V2, EfficientNetV2, and DarkNet-19—were adapted and fine-tuned through transfer learning. Deep feature representations were then extracted from each CNN model. These extracted features were subsequently integrated into a unified feature space using a sophisticated convolutional sparse image decomposition fusion technique. To further enhance the diagnostic accuracy, an entropy-controlled firefly algorithm was employed to meticulously select the most discriminative features. Finally, the optimally selected feature subset was fed into a robust multi-class support vector machine classifier for the final diagnosis of leukemia.

*Ansari et al. (2023)* provided a unique approach for extracting characteristics in a hierarchical and accurate manner in order to detect different forms of acute leukemia. This approach distinguishes between acute leukemia types, namely ALL and AML, by separating lymphocytes from monocytes.

# MATERIALS AND METHODS

## Dataset and preprocessing

For this study, a dataset consisting of 3,256 PBS images from 89 suspected ALL patients from *Aria et al. (2021)*, is used The Taleqani Hospital's bone marrow laboratory produced the images for this dataset (Tehran, Iran). This dataset comprises two classes benign and malignant. The benign class consists of hematogones PBS images which are quite similar to ALL. The malignant class consists of PBS images of ALL subtypes: Early-B, Pre-B, and Pro-B ALL. The DOI for the dataset repository is 10.34740/KAGGLE/DSV/2175623.

The preprocessing stage detailing the data augmentation techniques are included. These techniques expand the size and diversity of the training dataset. By creating modified versions of the existing images, more variability into the training process is introduced. This increased variability helps the model learn more robust and generalizable features, making it less susceptible to overfitting the specific characteristics of the original limited dataset. For instance, rotating an image of a blood cell allows the model to recognize the cell regardless of its orientation, while flipping exposes it to mirror images, further enhancing its ability to identify relevant features from different perspectives.

All the images in the dataset are resized to dimensions of $224 \times 224$ pixels with three color channels (RGB), resulting in a shape of $224 \times 224 \times 3$. This resizing ensures uniformity across the dataset, facilitating consistent input dimensions for the neural network. Additionally, the pixel values of each image are normalized to a range between 0 and 1. This normalization is achieved by dividing each pixel value by 255, as 255 is the maximum possible value for a pixel in an 8-bit image. This normalization step is crucial as it helps to improve the convergence speed and stability of the neural network during training, ensuring that the model learns more efficiently by keeping the input data within a consistent and manageable range. In the experiments, 80% of the total images have been chosen randomly for training, 10% for testing, and the remaining 10% for validation. Figures 1A–1D present several PBS images of Benign, Early-B, Pre-B, and Pro-B, respectively.

## Multi-task advanced convolutional neural network

Traditional machine learning often treats tasks in isolation. However, in real-world scenarios, tasks are frequently interconnected. For instance, in multi-task learning, where multiple objectives are learned simultaneously, the extent of information sharing between tasks can significantly impact performance. Rigid grouping of tasks can hinder learning, especially when task relationships are complex or dynamic. To address this, a novel approach is proposed, the Multi-Task Advanced Convolutional Neural Network (MTA-CNN). Unlike traditional CNNs, MTA-CNN employs a flexible architecture where the degree of connectivity between subnetworks varies. This dynamic connectivity enables the model to automatically identify and exploit relevant task relationships. By leveraging a gradient-based mechanism, the model learns to adjust the influence of supervisory signals across different layers, facilitating the transfer of knowledge between related tasks. This approach moves beyond simplistic binary categorizations of task relationships, allowing for a more nuanced understanding of task interdependence (*Rohani, Farsi & Mohamadzadeh, 2023*).

In specific scenarios, the proposed model exhibits specialized behavior. When connections within a group of related tasks are maximized $\alpha^{pq}(p = q)$ are 1 while connections between different groups are minimized (TTCs between groups approach 0), the model effectively reduces to a set of independent, single-task learners. Conversely, if all Task TTC factors $\alpha^{pq}(p, q \in T)$ are identical, the model simplifies into a standard,

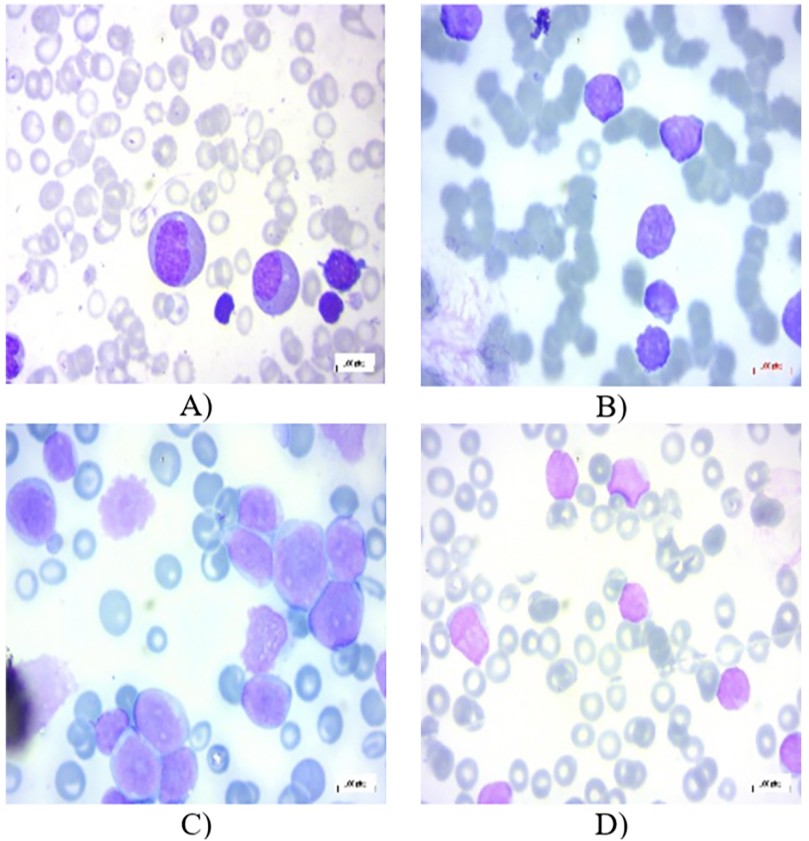

**Figure 1 Peripheral blood smear (PBS) images of (A) Benign (B) Early-B (C) Pre-B (D) Pro-B in the dataset.**

averaged multi-task CNN architecture. Furthermore, the proposed model can be shown to encompass the standard CNN model as a special case, specifically when the function $f(.)$ of weights $W$ and $A$ input adheres to the mathematical formulation outlined in Eq. (1).

$$f(A, W) \tag{1}$$

where $A$ represents the activation maps produced by the convolutional layers, $W$ denotes the weights of the convolutional filters. The primary objective is to minimize the $D$ which is the difference between the model's predicted output $B$ of $f(A, W)$ and the actual, observed values, as mathematically defined in Eq. (2).

$$min(D) \tag{2}$$

where $D$ is calculated as in Eq. (3).

$$D = -[B \ \ln(F) + (1 - B) \ \ln \ (1 - F)]. \tag{3}$$

A function of $A$, $W$, and TTC$^\alpha$ is utilized to define the multi-task advanced CNN method in the presence of $n$ tasks, represented as in Eq. (4).

$$F(A_1, A_2, \ldots, A_n, W_1, W_2, \ldots, W_n, \alpha). \tag{4}$$

Moreover, the multi-task method aims to reduce the cost of each task, is represented as in Eq. (5).

$$min\left(\sum_{p=1}^{n}(D_p)\right) \tag{5}$$

where $D_p$ is calculated as in Eq. (6), and Eq. (4) is updated as Eq. (7).

$$D_p = -\left[B_p \ ln \ (F_p) + (1 - B_p) \ ln \ (1 - F_p)\right] \tag{6}$$

$$F(A_p, W_1, W_2, \ldots, W_n, \alpha). \tag{7}$$

To constrain the TTC factor within a meaningful range, $\alpha$ value between 0 and 1, is enforced To achieve this, an auxiliary variable $\beta$ is introduced, defined by Eq. (8), which ensures the TTC factor remains within the desired bounds.

$$\alpha = sigmoid(\beta), \ \beta \in (-\infty, +\infty). \tag{8}$$

The core distinction between the multi-task model and conventional CNNs lies in the weight update mechanism. Unlike traditional CNNs, the model necessitates a more intricate weight adjustment process. This is because each subnetwork must simultaneously incorporate supervisory signals from both its own objectives and those of other interconnected subnetworks. While various optimization methods, such as Adaptive Gradient, RMSprop, and Stochastic Gradient Descent, offer distinct approaches to weight updates, a general formulation is presented here. The weight update rule for a standard CNN model can be formally expressed as shown in Eq. (9).

$$W_{t+1} = W_t + V_{t+1}. \tag{9}$$

The weight update process within the proposed framework is governed by updates of the CNN parameters. This process dynamically adjusts the weights $W^1, \ W^2, \ldots, W^n$ associated with all constituent elements $\alpha^{pq}$ of the TTC, including $n$ positions and associated parameters.

A variable, designated as $V_{t+1}^{W^{pq}}$, serves as an auxiliary element. This variable influences the update values within a specific optimization algorithm, particularly those associated with the method. The algorithm evaluates the cost incurred by the primary task. A critical parameter, denoted as $\beta^{pq}$, is introduced. This parameter modulates the influence of a secondary task on the primary task. Specifically, $\alpha_t^p$ acts as the $t + 1$ factor of $\beta^{pq}$ with respect to the primary task. This parameter effectively regulates the impact of the supervisory signal originating from the secondary task on the primary task. The algorithm iteratively updates the weights $W^p$ and $W_t^p$, represented by $t$ at iteration $t$ and $t + 1$, respectively. The value of the auxiliary variable, $V_{t+1}^{W^{pq}}$, is determined according to the mathematical expression defined in Eq. (10).

$$V_{t+1}^{W^{pq}} = \frac{\partial D_q}{\partial W_q} \cdot \frac{\partial W_q}{\partial W_p} \tag{10}$$

where $V_{t+1}^{\beta^{pq}}$ is calculated as in Eq. (11).

$$V_{t+1}^{\beta^{pq}} = \frac{\partial D_q}{\partial \alpha^{pq}} \cdot \frac{\partial \alpha^{pq}}{\partial \beta^{pq}}. \tag{11}$$

To prioritize the influence of the subnet's own objectives, the impact of supervisory signals from other subnets will be reduced. As a results, $\alpha_t^{pq}$ is set to 1 when $p$ equals $q$ ($p = q$). Batch normalization is employed to mitigate the volatility of feature value ranges. By using per-pixel class-weighted binary focal loss with Eq. (12), MTA-CNN can effectively optimize its binary predictions, especially in scenarios where class imbalance is a significant issue (*Lin et al., 2017*; *Dosovitskiy et al., 2020*).

$$l_{loss} = a_c y (1 - p_c)^{\gamma} \cdot \log p_c + (1 - y)^{\gamma} \cdot \log(1 - p_c) \tag{12}$$

where $y \in \{0,1\}$ is the ground-truth label, $p_c$ is the corresponding pixel-level estimation, $a_c = \frac{N_b}{N_f}$ is the class-weighting factor, and $\gamma$ is the focusing parameter. The class-weighting balances optimization for background and foreground, indirectly aiding contour estimation. Focal loss prioritizes harder-to-classify pixels, improving overall performance.

The proposed model outperforms traditional training methods by enabling incremental task addition while ensuring the integrity of each task's subnetwork. As training advances, subnetworks for new tasks form connections with previously learned tasks, acquiring unique parameters. Crucially, the parameters of subnetworks for already learned tasks remain fixed. In this scenario, knowledge acquired from two previously trained tasks is leveraged to facilitate the learning of a new, distinct task. Figure 2 shows the proposed scheme with the proposed multi-task advanced CNN based network with both classification and segmentation procedures.

The primary task is hierarchical classification. In this task, first it is decided whether it is leukemic or not, and then if it is leukemic, which subtype it belongs to. The secondary task is nucleus segmentation: Automatically segmenting the nuclei of leukemic cells.

*Training task:* These are the tasks used in the initial training of the model. Usually, the basic classification task (4-class classification) and one or more secondary tasks that can help, nucleus segmentation, are combined. Figure 3 presents the proposed MTA-CNN architecture.

*New task*: After training, a new task is added to the model. In a trained model, the cell nucleus segmentation task is added as the task of automatically segmenting the nuclei of leukemic cells.

After training as new task, a new task is added to the model. In a trained model, a cell nucleus segmentation task is added. This segmentation performs segmentation with a new TransUnet. A TransuNet-based architecture is proposed as the segmentation process. To identify and delineate lesions within medical images, a TransUnet segmentation model is employed. This approach leverages the strengths of both Vision Transformers (ViTs) and traditional U-Nets. In the experiments, the 100 heads are used, each with a feature dimension of four within the TransUNet's ViT component.
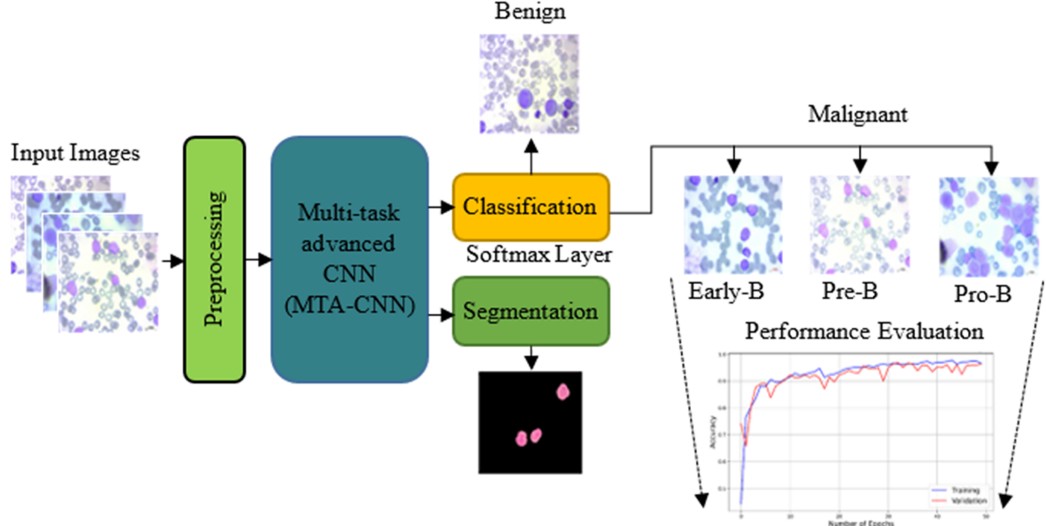

**Figure 2 The proposed scheme with the proposed multi-task advanced CNN based network.**

Patch Embedding

Input Image: An image of size $H \times W \times C$ (height, width, channels).

Patch Size: A fixed size $P \times P$ patch.

Number of Patches: $N = (H \times W)/(P \times P)$.

The image is divided into $N$ non-overlapping patches. Each patch is flattened and projected into a D-dimensional token:

$$x_i = \left[ x_i^1, x_i^2, \ldots, x_i^D \right] = f(p_i) \tag{13}$$

where $x_i$ is the $i$-th patch token, $p_i$ is The $i$-th image patch, $f$ is A linear projection function.

Positional encoding is added to the token embeddings to provide information about the relative or absolute position of the patch within the image.

$$x_i = x_i + PE(i) \tag{14}$$

where $PE(i)$ is Positional encoding for the $i$-th token, the attention score between two tokens $i$ and $j$ is calculated as:

$$Attention(Q, K, V) = Softmax\left(Q. K^T/sqrt(d_k)\right) V \tag{15}$$

where $Q, K, V$ are Matrices of queries, keys, and values, respectively, $d_k$ is the dimension of the key vectors.

*Multi-head self-attention*: Multiple self-attention layers are used in parallel to capture different aspects of the input sequence. The outputs of these layers are concatenated and linearly projected to produce the final output.

*Feed-forward neural network*: A simple feed-forward neural network with two fully-connected layers is applied to each token independently.

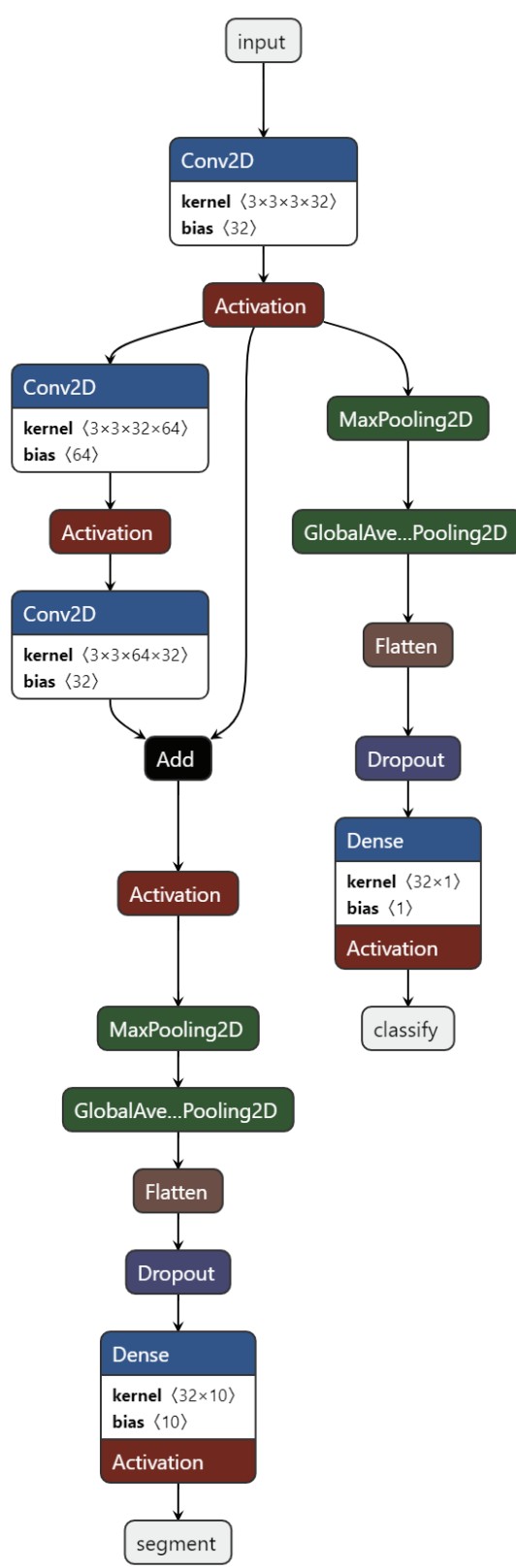

**Figure 3** **The proposed MTA-CNN architecture.**

*Layer normalization*: Layer normalization is used to stabilize the training process. It normalizes the activations of each layer.

## RESULTS

This article proposes a novel MTA-CNN for robust ALL diagnosis. The MTA-CNN integrates three critical tasks: diagnosis, classification, and segmentation of ALL cells within a single framework. The performance analysis of the proposed method was performed with the models explained below and comparison was made on various models.

*EfficientNetB0*: A family of models that achieve state-of-the-art performance with fewer parameters and computations, making them suitable for resource-constrained devices (*Bhuma & Kongara, 2020*).

*Vision Transformer (ViT)* is a deep learning model that has revolutionized the field of image processing. This model was inspired by the Transformer architecture, a breakthrough in natural language processing as demonstrated by its remarkable success in various applications (*Chen et al., 2021*; *Dosovitskiy et al., 2020*).

*TransUNet*: TransUNet is a hybrid architecture that combines the strengths of CNNs and Transformers (*Chen et al., 2021*). It typically consists of a Transformer Encoder and a U-Net Decoder. The Transformer Encoder processes the input image using multiple Transformer blocks, each composed of self-attention and feed-forward neural network layers.

*SegFormer*: SegFormer is a hybrid architecture that combines the strengths of CNNs and Transformers (*Xie et al., 2021*). It consists of a hierarchical Transformer encoder and a pyramid pooling module.

*Swin Transformer*: Swin Transformer is a hierarchical Transformer-based vision model that excels in capturing long-range dependencies at different scales (*Liu et al., 2021*). It employs a hierarchical structure of Transformer blocks, where each block processes a progressively smaller image resolution.

*BoundaryNet*: A novel approach for medical image segmentation that focuses on accurately predicting boundaries between different objects or regions (*Trivedi & Sarvadevabhatla, 2021*).

## EXPERIMENTAL SETUP

The proposed method and other models have been conducted on Windows 10 operating system installed on a computer having an Intel CoreTM i7-8700 processor, 16 GB RAM, and an Nvidia Geforce 4 GB Graphics Card device. All models in the experiments are coded using Python 3.8.5 programming. Keras and Tensorflow libraries are utilized in the programs for training the proposed networks. Table 1 presents several significant parameters used in experiments for this study. To use the memory effectively, the batch size is adjusted to 32 for the 2-D CNN model. Here, a batch size includes 32 slices used for each training iteration.

**Table 1  The significant parameters used in the experiments.**

| Parameters | Definition |
| --- | --- |
| Kernel size of the convolution layer | $(3 \times 3)$ kernel size is used |
| Output | Classification with four classes (Benign, Early-B, Pre-B, and Pro-B) |
| Learning rate | 0.001 |
| Optimization method | Adam |
| Batch size | 32 |
| Number of epochs | 50 |
| Dropout | 0.2 |

To analysis the proposed and other methods, several evaluation metrics are used. Note that true positives ($TP$) are correctly identified positives, true negatives ($TN$) are correctly identified negatives, false positives ($FP$) are incorrectly identified positives, and false negatives ($FN$) are incorrectly identified negatives.

Precision ($P_{rc}$) is a measure of how many of all the positive predictions a classification model makes are actually positive. It is calculated as in Eq. (16).

$$P_{rc} = \frac{TP}{TP + FP}.$$
(16)

Recall ($R_{call}$) is a measure of a model's ability to correctly identify all relevant instances of a class. It measures the proportion of actual positive cases that were correctly classified as positive by the model as given in Eq. (17).

$$R_{call} = \frac{TP}{TP + FN}.$$
(17)

F1-score ($F_{scr}$) is a harmonic mean of precision and recall. It provides a single metric that balances both precision and recall, as calculated in Eq. (18).

$$F_{scr} = 2 \times \frac{P_{rc} \times R_{call}}{P_{rc} + R_{call}}.$$
(18)

Specificity ($S_{ty}$) is a statistical measure that indicates the proportion of true negatives identified by a test. It measures how well a test can correctly identify individuals who do not have a particular condition, given in Eq. (19).

$$S_{ty} = \frac{TN}{TN + FP}.$$
(19)

Cohen's Kappa is a statistical measure of inter-rater agreement for categorical data. It corrects for the chance agreement that might occur by chance alone, providing a more accurate assessment of the extent to which two or more raters (or a model and ground truth) agree on their judgments. A Kappa coefficient of 1 indicates perfect agreement, while a Kappa of 0 indicates no agreement beyond what would be expected by chance. Cohen's Kappa is calculated as in Eq. (20). Two specialists raters have taken into account in the experiments

$$C_{kap} = \frac{P(a) - P(e)}{1 - P(e)} \tag{20}$$

where $P(a)$ and P(e) are probability of agreement and probability of agreement by chance defines in Eqs. (21) and (22), respectively.

$$P(a) = \frac{TP + TN}{T_{ob}} \tag{21}$$

$$P(e) = \frac{TP * TN + FP * FN}{T_{ob}^2} \tag{22}$$

where $T_{obs}$ is the total observations ($TP + TN + FP + FN$), it is the total number of items rated by both raters.

Positive predictive value (PPV) and negative predictive value (NPV) are statistical measures used in diagnostic testing. They indicate the probability that a test result is correct, given the actual presence or absence of a condition. The NPV calculation is only given as in Eq. (23) since PPV is equal to precision.

$$NPV = \frac{TN}{TN + FN}. \tag{23}$$

Accuracy ($A_{cc}$) is the overall correct predictions. It measures the proportion of both positive and negative correct classifications out of the total number of observations. $A_{cc}$ is calculated as in Eq. (24).

$$A_{cc} = \frac{TP + TN}{T_{obs}}. \tag{24}$$

## The performance results of the ALL diagnosis and classification

This section discusses and analyzes the results of the ALL diagnosis and classification obtained from PBS images. The classification results as Benign, Early-B, Pre-B, and Pro-B in an image from images are evaluated.

As seen from Fig. 4, throughout the training and optimization cycles of the proposed hybrid CNN, notable enhancements in performance were observed. As training progressed with more epochs, the frequency of weight adjustments within the neural network diminished, leading to consistent convergence and improved model fitting. This process also led to a corresponding increase in overall accuracy (see Fig. 4A) and a reduction in the loss function (see Fig. 4B), contributing positively to the model's performance.

Table 2 shows the model's performance on both training and validation datasets across different classes of ALL cells (Benign, Early-B, Pre-B, and Pro-B). The model exhibits high precision, recall, F1-score, and accuracy values across all classes in both training and validation sets. This suggests that the model has learned the underlying patterns in the data effectively and can generalize well to unseen data. The consistent high performance across all classes indicates the model's ability to accurately distinguish between different types of ALL. While the performance on the training set is slightly higher than on the validation set, the difference is minimal, suggesting that the model is not overfitting to the training data.
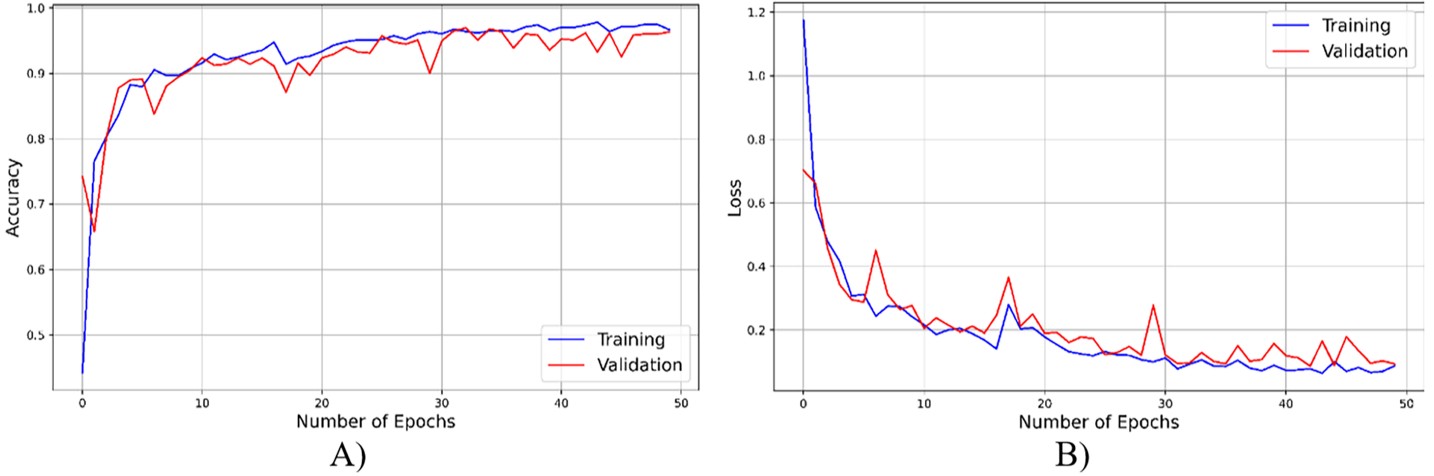

**Figure 4 The accuracy result of the proposed model.**

**Table 2 The performance results of the proposed model.**

| Type | Classes | | Precision | Recall | F1-score | Accuracy (%) |
|------|---------|---|-----------|--------|----------|--------------|
| Training | Benign | | 0.989 | 0.940 | 0.964 | 98.924 |
| | Malignant | Early-B | 0.985 | 0.995 | 0.990 | |
| | | Pre-B | 0.994 | 1.000 | 0.997 | |
| | | Pro-B | 0.972 | 1.000 | 0.986 | |
| Validation | Benign | | 0.967 | 0.900 | 0.932 | 97.695 |
| | Malignant | Early-B | 0.976 | 0.985 | 0.980 | |
| | | Pre-B | 0.984 | 0.989 | 0.986 | |
| | | Pro-B | 0.979 | 1.000 | 0.989 | |

The results demonstrate the model's effectiveness in accurately diagnosing and classifying ALL cells, making it a promising tool for improving the efficiency and accuracy of ALL diagnosis and treatment planning.

Figure 5 shows the confusion matrix of the model for both training and validation.

Table 3, which compares the performance of different learning methods for ALL diagnosis. The table presents various performance metrics, including accuracy, precision, recall, F1-score, specificity, Cohen's Kappa, and negative predictive value (NPV) for each model: EfficientNetB0, ViT, TransUNet, SegFormer, Swin Transformer, and BoundaryNet. All models demonstrate high accuracy, with BoundaryNet achieving the highest accuracy at 0.9735. Precision and recall values are consistently high across all models, indicating a low rate of false positives and false negatives in identifying ALL cells. F1-scores, which balance precision and recall, are also high, suggesting that the models effectively identify and classify ALL cells. Specificity, which measures the ability to correctly identify non-ALL cells, is also high across all models. Cohen's Kappa and NPV further support the models' strong performance, indicating high agreement between predictions and ground truth and high confidence in negative predictions.

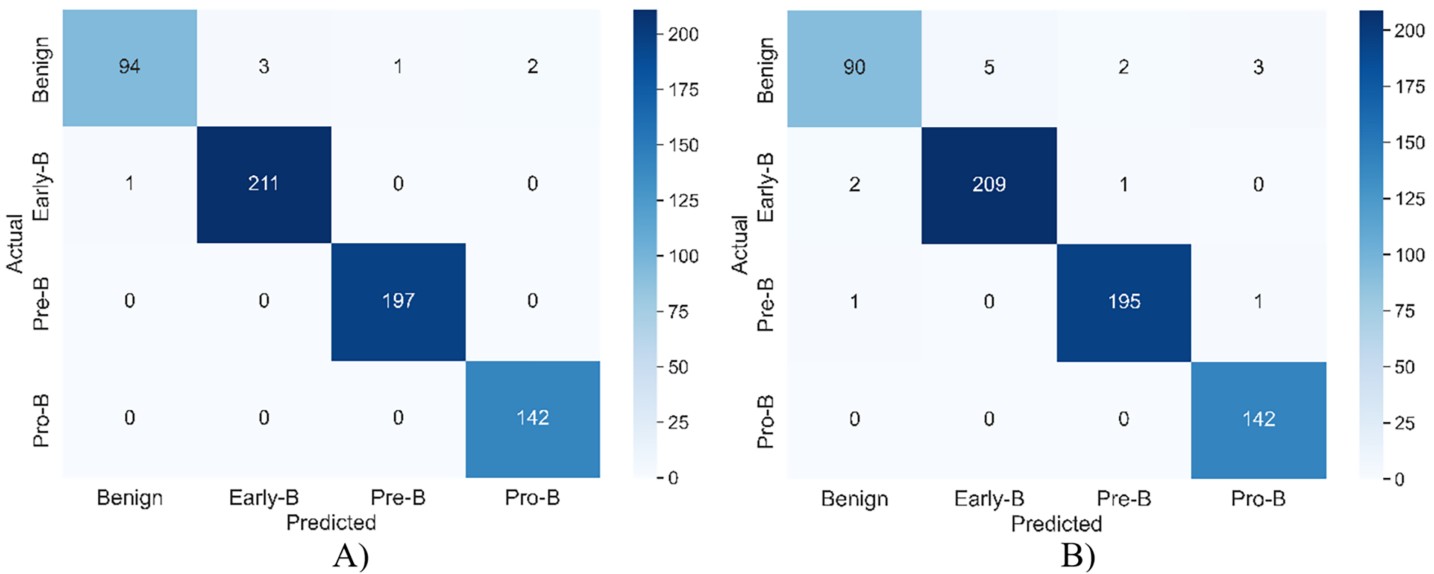

**Figure 5** The confusion matrix of the model for (A) training (B) validation. 

**Table 3 The performance comparison results of the different learning methods.**

| Model | Accuracy (%) | Precision | Recall | F1-score | Specificity | Cohen's Kappa | NPV |
|---|---|---|---|---|---|---|---|
| EfficientNetB0 (*Bhuma & Kongara, 2020*) | 95.84 | 0.963 | 0.955 | 0.959 | 0.982 | 0.943 | 0.984 |
| Vision Transformer (ViT) (*Chen et al., 2021*) | 96.23 | 0.971 | 0.958 | 0.965 | 0.987 | 0.951 | 0.986 |
| TransUNet (*Chen et al., 2021*) | 96.46 | 0.967 | 0.957 | 0.962 | 0.985 | 0.944 | 0.983 |
| SegFormer (*Xie et al., 2021*) | 96.52 | 0.975 | 0.963 | 0.969 | 0.989 | 0.955 | 0.988 |
| Swin Transformer (*Liu et al., 2021*) | 96.63 | 0.973 | 0.962 | 0.968 | 0.988 | 0.943 | 0.987 |
| BoundaryNet (*Trivedi & Sarvadevabhatla, 2021*) | 97.35 | 0.978 | 0.965 | 0.972 | 0.990 | 0.952 | 0.989 |
| MTA-CNN (proposed) | 97.88 | 0.979 | 0.967 | 0.973 | 0.991 | 0.979 | 0.990 |

## The performance results of the ALL segmentation

The segmentation stages of lymphoblastic leukemia blood cells is analyzed in Figs. 6, 7, and 8, and evaluate the performance of the proposed MTA-CNN method. Figure 6 illustrates the segmentation process for a malignant Early-B blood smear image. The original image (a) shows a mix of healthy and cancerous cells. The initial segmentation (b) identifies some cancerous cells but also misclassifies healthy cells and misses some cancerous ones. However, applying the MTA-CNN method significantly improves the segmentation (c). In the final image, cancerous cells are clearly defined, and the number of incorrectly classified cells (both false positives and negatives) is substantially reduced.

Figure 7 depicts the segmentation process for a malignant Pre-B blood smear image. Like Fig. 6, it begins with a raw image (a) containing a mixture of healthy and cancerous cells. However, the initial segmentation result (b) in this case exhibits a lower accuracy compared to Fig. 6. This is evident from the presence of larger areas where the algorithm

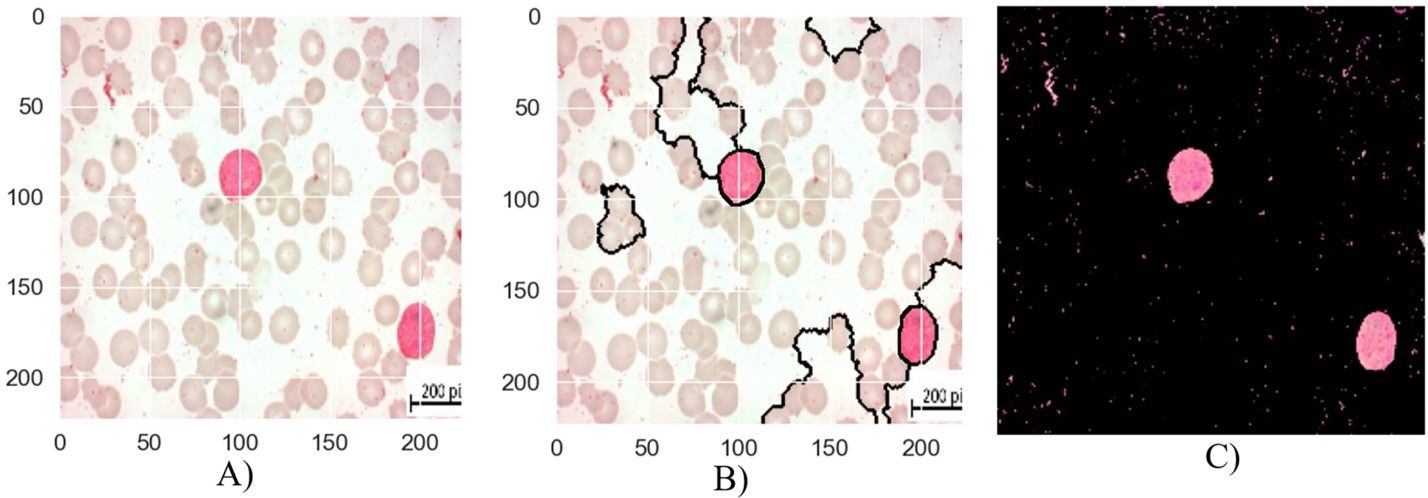

**Figure 6 The results of the segmentation of a malignant Early-B image (A) original image (B) masked and segmented image (C) improved segmented image.**

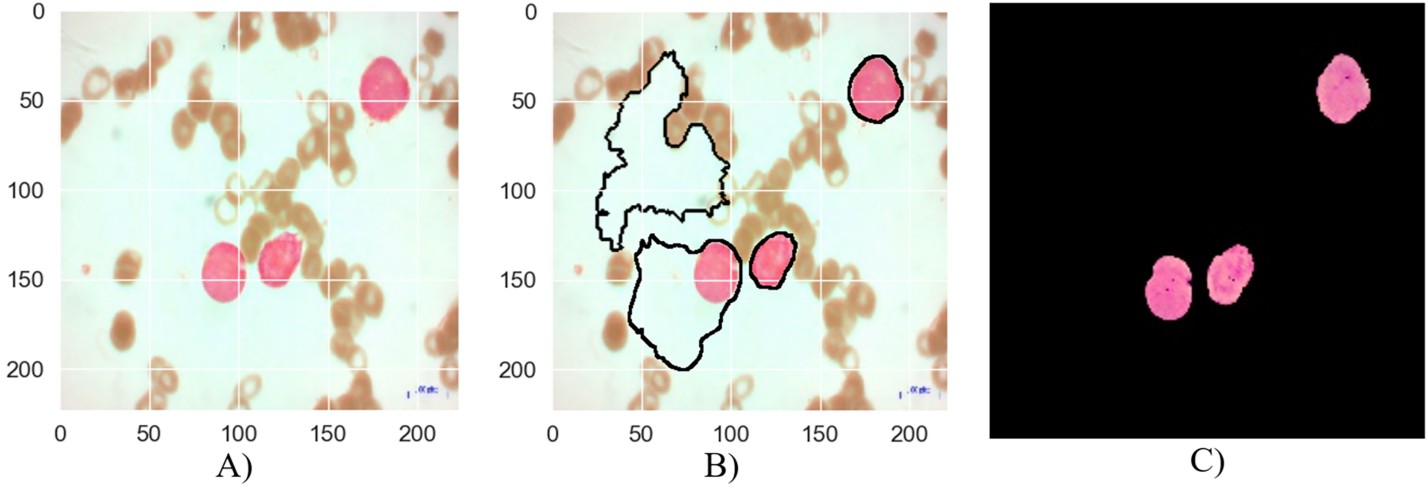

**Figure 7 The results of the segmentation of a malignant Pre-B image (A) original image (B) masked and segmented image (C) improved segmented image.**

incorrectly classifies cells, either identifying healthy cells as cancerous or failing to detect cancerous cells entirely. The application of the MTA-CNN method significantly enhances the segmentation outcome (c). The improved segmentation demonstrates a marked improvement with cancerous cells being clearly delineated and effectively separated from the surrounding background. This refined segmentation provides a more accurate representation of the cancerous cell distribution within the blood smear.

Figure 8 presents the segmentation results for a malignant Pro-B blood smear image. The original image (a) depicts the raw blood smear sample. The initial segmentation attempt (b) demonstrates some inaccuracies, with certain cancerous cells not fully encompassed within the segmentation boundaries, while some healthy cells are

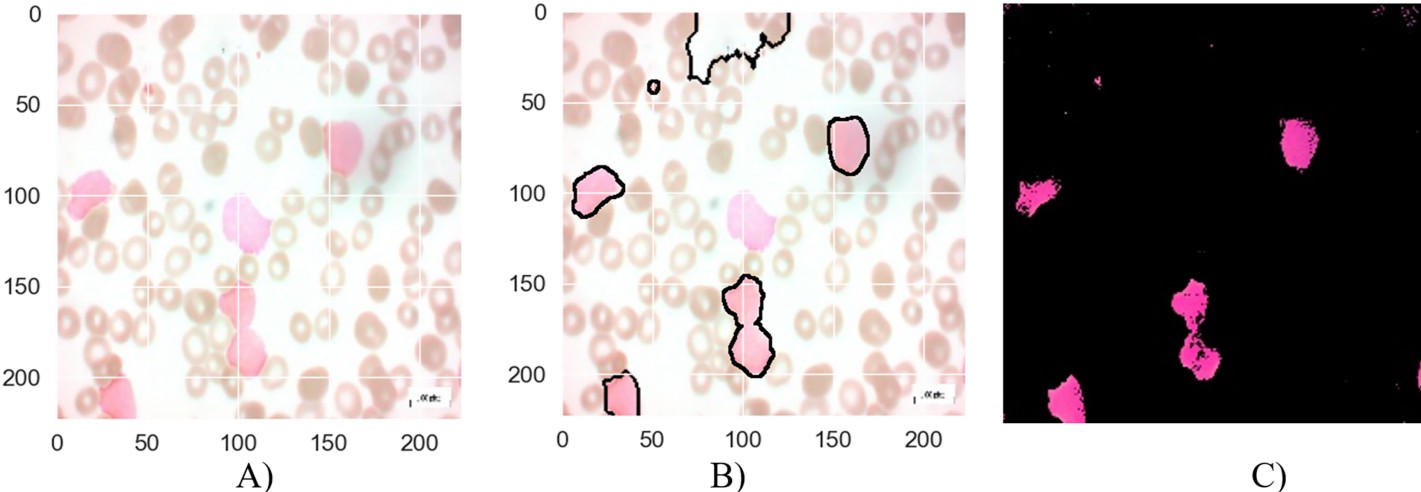

**Figure 8 The results of the segmentation of a malignant Pro-B image (A) original image (B) masked and segmented image (C) improved segmented image.**

erroneously classified as cancerous. The application of the MTA-CNN method significantly enhances the segmentation accuracy (c). This refined segmentation provides a more precise delineation of the cancerous cells, effectively isolating them within the segmented regions and reducing the instances of misclassification.

Quantitatively evaluating the performance of segmentation requires metrics are:

- Dice similarity coefficient (DSC): Measures the overlap between the ground truth segmentation and the predicted segmentation. Higher DSC values indicate better performance.
- Jaccard Index (IoU): Similar to DSC, it measures the overlap between the two segmentations.

Table 4 presents the segmentation performance results of all the algorithms compared. In Table 4, the performances of different image segmentation algorithms are compared according to DSC and IoU metrics. These metrics are frequently used to measure the accuracy of segmentation results. DSC expresses the similarity between two clusters, and IoU expresses the ratio of the intersection area to the union area. In the table, it is seen that MTA-CNN algorithm obtains the highest or closest value in both DSC and IoU metrics. This shows that MTA-CNN performs the segmentation task better than other algorithms. In particular, a high DSC value indicates that the segmentation masks created by MTA-CNN are more similar to the real masks, *i.e.*, it provides a more accurate segmentation. A high IoU value indicates that the overlap between the segmentation masks and the real masks is greater. When looking at other algorithms, it is seen that popular models such as EfficientNetB0, Vision Transformer, TransUNet, SegFormer, Swin Transformer and BoundaryNet also perform well. However, the DSC and IoU values of these models are generally below MTA-CNN. This may be due to the fact that MTA-CNN has an architecture specifically designed for the segmentation task.

**Table 4  The segmentation performance of the all compared algorithms.**

| Algorithms | DSC (Average) | IoU (Average) | F1-score | Training time (approximately h) | Pre-evaluation |
|---|---|---|---|---|---|
| EfficientNetB0 (*Bhuma & Kongara, 2020*) | 0.88 | 0.80 | 0.90 | 8.2 | Good baseline performance, but inferior to MTA-CNN |
| Vision Transformer (ViT) (*Chen et al., 2021*) | 0.91 | 0.84 | 0.92 | 12.4 | Strong performance can deliver good results, especially on large datasets |
| TransUNet (*Chen et al., 2021*) | 0.93 | 0.86 | 0.93 | 15.6 | A model that gives good results in medical imaging |
| SegFormer (*Xie et al., 2021*) | 0.90 | 0.83 | 0.91 | 11.4 | A fast and efficient model, but may perform less well in some situations than others |
| Swin Transformer (*Liu et al., 2021*) | 0.92 | 0.85 | 0.92 | 14.3 | Can give good results for high resolution images |
| BoundaryNet (*Trivedi & Sarvadevabhatla, 2021*) | 0.89 | 0.82 | 0.91 | 9.7 | A model that performs well in identifying boundaries |
| **MTA-CNN (Proposed)** | **0.95** | **0.90** | **0.96** | **12.1** | **Top performer, leader in all metrics** |

# CONCLUSION

In this study, a novel multi-task advanced convolutional neural network, is presented specifically designed to address the critical challenges associated with accurate and robust ALL diagnosis, classification, and nucleus segmentation from PBS images. The proposed MTA-CNN leverages a shared encoder to extract discriminative features, which are then fed into task-specific decoders for classification and segmentation. This multi-task learning approach enables the model to learn more comprehensive and robust representations, resulting in improved performance across all tasks. The experimental results, conducted on a challenging dataset of PBS images, demonstrated the superior performance of the MTA-CNN compared to several state-of-the-art methods. The proposed model achieved significantly higher accuracy in classifying ALL subtypes into the benign, Early-B, Pre-B, and Pro-B, segmenting nuclei, and distinguishing between benign and malignant cells. These improvements are attributed to the model's ability to effectively capture complex image patterns and handle variations in image quality. The successful application of the MTA-CNN to ALL diagnosis, classification, and segmentation has the potential to revolutionize the field of hematopathology. By automating these tasks, the proposed model can significantly reduce the workload of pathologists, improve diagnostic accuracy, and enable earlier and more precise treatment planning. This can ultimately lead to better patient outcomes and improved survival rates. While the proposed model has shown promising results, there are still opportunities for further improvement. Future research may explore the integration of attention mechanisms to enhance feature extraction, the incorporation of 3D CNNs to handle volumetric data, and the development of more efficient and lightweight models for deployment on resource-constrained devices. Additionally, investigating the impact of data augmentation techniques and transfer learning on model performance could be a fruitful direction. In conclusion, this study has demonstrated the effectiveness of deep learning techniques in addressing the challenges of ALL diagnosis, classification, and segmentation. The proposed MTA-CNN offers a

powerful tool for clinical decision-making and has the potential to significantly impact patient care. However, continued research is needed to address the limitations and further enhance the model's performance, generalization, and clinical applicability.

### Funding

The authors received no funding for this work.

### Competing Interests

Bilal Alatas is an Academic Editor for PeerJ.

### Author Contributions

- Sercan Yalcin conceived and designed the experiments, performed the experiments, analyzed the data, performed the computation work, prepared figures and/or tables, authored or reviewed drafts of the article, and approved the final draft.
- Zuhal Cetin Yalcin conceived and designed the experiments, analyzed the data, prepared figures and/or tables, and approved the final draft.
- Muhammed Yildirim conceived and designed the experiments, performed the experiments, analyzed the data, performed the computation work, authored or reviewed drafts of the article, and approved the final draft.
- Bilal Alatas conceived and designed the experiments, authored or reviewed drafts of the article, and approved the final draft.

### Data Availability

The codes are available in the Supplemental File.

The Acute Lymphoblastic Leukemia (ALL) image dataset is available at Kaggle:

- Mehrad Aria, Mustafa Ghaderzadeh, Davood Bashash, Hassan Abolghasemi, Farkhondeh Asadi, and Azamossadat Hosseini. (2021). Acute Lymphoblastic Leukemia (ALL) image dataset [Data set]. Kaggle. https://doi.org/10.34740/KAGGLE/DSV/2175623.

### Supplemental Information

Supplemental information for this article can be found online at http://dx.doi.org/10.7717/peerj-cs.3043#supplemental-information.

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
