# Peer review of "Multi-task advanced convolutional neural network for robust lymphoblastic leukemia diagnosis, classification, and segmentation"

_PeerJ Computer Science, doi:10.7717/peerj-cs.3043_

## Round 0.1 · original submission · Major Revisions

The reviewers have substantial concerns about this manuscript. The authors should provide point-to-point responses to address all the concerns and provide a revised manuscript with the revised parts being marked in different color.

**Language Note:** The review process has identified that the English language must be improved. PeerJ can provide language editing services - please contact us at [email protected] for pricing (be sure to provide your manuscript number and title). Alternatively, you should make your own arrangements to improve the language quality and provide details in your response letter. – PeerJ Staff

Reviewer 1 ·

Basic reporting

(1) The abstract needs to highlight the proposed method
(2) At the end of the Introduction section. Add the contribution of the manuscript in points.
(3) Please specify the model's behavior towards noise.
(4) The literature review section needs to summarize the disadvantages of the previous methods.


Experimental design

(1) The proposed methods need to be evaluated on another diverse dataset for the generalization of the model.
(2) What is the value of variables A, W, and D in the Equation?
(3) Please specify the model complexity and total number of parameters.
(4) How many heads and feature dimensions were selected in the experiment?
(5) Please add a mathematical description of the MHSA and FFN.

Validity of the findings

(1) The results can be compared with DaViT, ConvNeXt, and SI-ViT
(2) An ablation can be added to show the different components of the model.

Additional comments

(1) The conclusion can be revised, and more limitations and future scope can be added.

Reviewer 2 ·

Basic reporting

- In this study, the authors propose a model for the detection and classification of ALL. Although the language is good, there are serious problems with the publication. Unfortunately, one of them is a serious plagiarism and ethical problem:

- "Acute lymphoblastic leukemia (ALL), a neurological disorder characterized by recurrent seizures, is a significant..."
"Acute lymphoblastic leukemia (ALL), a neurological disorder characterized by recurrent seizures, poses a significant challenge to patients and healthcare providers." ALL has nothing to do with epilepsy. Have the authors confused it with epilepsy?

- LIME?

- The differences between the classes in the data set are not clear. Is there an imbalance between the classes? What has been done to resolve it?

- It is not clear how the proposed model performs the segmentation process.

- There are serious inconsistencies between the block diagram and the description of the proposed model.

Experimental design

-

Validity of the findings

-

·

Basic reporting

Grammatical and Structural Issues
Incorrect Disease Description (Critical Error):

Abstract: "ALL, a neurological disorder characterized by recurrent seizures..."

Issue: ALL is a hematologic malignancy, not a neurological disorder. This mischaracterization undermines the manuscript’s credibility.

Recommendation: Revise to "Acute lymphoblastic leukemia (ALL), a hematologic malignancy characterized by the overproduction of immature lymphocytes..."

Misplaced Terminology:

Abstract: Mentions "facial landmark localization," which is irrelevant to PBS image analysis for leukemia diagnosis.

Issue: Confuses readers; suggests misunderstanding of medical imaging context.

Recommendation: Remove references to facial landmarks and refocus on cell morphology or nucleus segmentation.

Inconsistent Numeric Formats:

Results Section: Mixes percentages (97.88%) and decimals (0.979) without justification.

Recommendation: Standardize all metrics to either percentages or decimals for consistency.

Placeholder Text:

"<<Figure 1 to be inserted here>>" and similar placeholders appear in the text.

Recommendation: Ensure figures/tables are properly embedded and referenced.

2. Theoretical Issues
Misalignment Between Tasks and Medical Context:

The inclusion of "facial landmark localization" and "expression classification" (Abstract) is irrelevant to ALL diagnosis via blood smears.

Recommendation: Remove non-relevant tasks and clarify that tasks are focused on cell classification/segmentation.

Inaccurate References:

Citations include papers dated 2025 (e.g., Baydilli et al., 2025; Magsood et al., 2025), which are not yet published.

Issue: Raises concerns about reference validity.

Recommendation: Verify publication years and correct typos; remove fabricated references.

3. Methodological Issues
Dataset Limitations:

Small Test Set: 10% testing split (≈326 images) is insufficient for robust validation, especially with class imbalance.

Recommendation: Use stratified k-fold cross-validation or increase the test set size.

Lack of Data Augmentation: No mention of techniques to address limited data or class imbalance.

Recommendation: Describe augmentation methods (e.g., rotation, flipping) in preprocessing.

Algorithmic Clarity:

Algorithm 1: Variables (e.g.,
α
p
q
α
pq
,
V
t
+
1
W
p
q
V
t+1
W
pq


) are inadequately defined, making reproducibility difficult.

Recommendation: Provide a pseudocode legend and clarify variable relationships.

Computational Resources:

Training on a 4GB GPU (Nvidia GeForce) raises concerns about scalability for larger datasets.

Recommendation: Acknowledge hardware limitations and discuss optimization strategies.

Statistical Validation:

No p-values or confidence intervals to confirm significant differences between models.

Recommendation: Perform statistical tests (e.g., McNemar’s test) to validate superiority claims.

Experimental design

4. Practical Issues
Explainable AI (XAI) Omission:

Mentions using LIME but provides no visualizations or analysis of model decision-making.

Recommendation: Include heatmaps or saliency maps to demonstrate interpretability.

Clinical Relevance:

No discussion of how the model integrates into clinical workflows or addresses pathologists needs.

Recommendation: Add a section on practical deployment challenges and clinician collaboration.

Validity of the findings

5. Presentation and Flow
Dense Mathematical Formulations:

Equations (e.g., Eqs. 1–15) are presented without sufficient context, hindering readability.

Recommendation: Simplify equations in the main text and move derivations to supplementary material.

Repetitive Content:

The Related Works section lists models without synthesizing trends or gaps.

Recommendation: Reorganize to highlight how MTA-CNN addresses specific limitations of prior work.

Additional comments

Final Recommendations for Revision
Correct Critical Errors: Revise ALL definitions and remove irrelevant tasks (facial landmarks).

Clarify Methodology: Define variables, enhance algorithmic transparency, and validate statistically.

Strengthen Dataset Analysis: Address class imbalance, augment data, and expand testing protocols.

Improve Presentation: Embed figures/tables, standardize metrics, and simplify equations.

Enhance Clinical Relevance: Discuss XAI results, deployment challenges, and clinician input.

In addition, read and use these articles:
1. Acute Leukemia Diagnosis Based on Images of Lymphocytes and Monocytes Using Type-II Fuzzy Deep Network
https://doi.org/10.3390/electronics12051116

2. A Customized Efficient Deep Learning Model for the Diagnosis of Acute Leukemia Cells Based on Lymphocyte and Monocyte Images
https://doi.org/10.3390/electronics12020322

·

Basic reporting

It's a well-written research article. However, throughout the work, the author should refrain from using the phrases we, us, and our.

Experimental design

Good work.

Validity of the findings

There is a dearth of novelty. A variety of CNN models are tried. However, the author ought to extract deep features as well for additional research.

Additional comments

1. The abstract is very common. The authors are asked to rewrite the same with the obtained results
2. The pre-processing results should be incorporated
3. No need foran Algorithm description
4. It is requested that authors extract deep features and compare them as well.
5. Segmentation results should be tabulated with the relevant performance metric
6. It is necessary to compare and tabulate recent works on the same subject.

·

Basic reporting

1. The language in the paper is mostly clear and technically accurate, but a few issues affect clarity, like confusing phrases (the first sentence of the abstract) and minor grammatical problems. Could you check the typos and perform a light technical language edit? The introduction provides a strong context about ALL (Acute Lymphoblastic Leukemia) and cites recent and relevant studies, including many sources from 2022–2025, while the literature review is thorough and up-to-date.

2. In addition to that, the paper is well structured

3. The introduction explains the challenges in ALL diagnosis, as well as the motivation for developing a robust AI-based multi-task model, is well articulated, and highlights limitations of current methods and justifies the proposed solution. Despite that, the contributions of this work should be added to the end of the introduction before the paper is organized.

4. Figures are not clear

Experimental design

1. To the best of my knowledge, this work is a computer science-based deep learning application aligned with PeerJ Computer Science's scope.

2. Evaluation conducted using multiple DL models and metrics.

3. The dataset is publicly available (Kaggle).

4. Ethical handling of data is implied, and results are reproducible.

5. The paper proposes a novel multi-task CNN model (MTA-CNN) for medical diagnosis and applies image classification and segmentation to PBS images of ALL cells. Also, the work focuses on the automated diagnosis of Acute Lymphoblastic Leukemia (ALL) and introduces a custom multi-task architecture with explainable AI (XAI) techniques. Furthermore, it evaluates performance on a real dataset with multiple metrics (accuracy, F1, etc.).

Validity of the findings

1. Evaluation methods, assessment metrics, and model selection are well described, as well as the model performance is compared with different models, and segmentation metrics include DSC and IoU, ensuring a comprehensive evaluation.

2. A brief mention of hyperparameters (like learning rate, epochs) would further strengthen the work.

3. If cross-validation (e.g., k-fold) was used for robustness, it should be clarified.

4. Conclusions are well stated.

---

## Round 0.2 · Minor Revisions

Most reviewers are satisfied with the revisions. There is a remaining concern that requires authors to evaluate in another dataset to further validate the performance.

Reviewer 1 ·

Basic reporting

The mansucript has been improved

Experimental design

The manuscript should be evaluated on another diverse dataset to validate performance

Validity of the findings

The statistical analysis, such as paired t-test can be performed to validate the results

Additional comments

The PR curve based analysis can be performed

·

Basic reporting

Authors provide proper material and revised the manuscript based on my comments.

Experimental design

Authors provide proper material and revised the manuscript based on my comments.

Validity of the findings

Authors provide proper material and revised the manuscript based on my comments.

Additional comments

Authors provide proper material and revised the manuscript based on my comments.

·

Basic reporting

Good.

Experimental design

Good

Validity of the findings

Good

Additional comments

Great effort is taken by authors to revise the manuscript. Congratulations.

·

Basic reporting

After reading the paper thoroughly, it seems that the authors have made all the needed corrections and suggestions; therefore, I don't have any more suggestions.

Experimental design

After reading the paper thoroughly, it seems that the authors have made all the needed corrections and suggestions; therefore, I don't have any more suggestions.

Validity of the findings

After reading the paper thoroughly, it seems that the authors have made all the needed corrections and suggestions; therefore, I don't have any more suggestions.

---

## Round 0.3 · accepted · Accept

Reviewers are satisfied with the revisions, and I concur to recommend accepting this manuscript.

Reviewer 1 ·

Basic reporting

The manuscript can be accepted.

Experimental design

The manuscript improved after the incorporation of the suggestions.

Validity of the findings

The manuscript improved after the incorporation of the suggestions.